# Non-contact lower limb injuries in Rugby Union: A two-year pattern recognition analysis of injury risk factors

**Seren Lois Evans**[1]*, **Robin Owen**[2], **Gareth Whittaker**[3], **Oran Elphinstone Davis**[3], **Eleri Sian Jones**[4], **James Hardy**[4], **Julian Owen**[1]

**1** Institute for Applied Human Physiology, School of Human and Behavioural Sciences, Bangor University, Bangor, United Kingdom, **2** School of Health and Sport Sciences, Liverpool Hope University, Liverpool, United Kingdom, **3** Welsh Rugby Union, Colwyn Bay, United Kingdom, **4** Institute for Psychology of Elite Performance, School of Human and Behavioural Sciences, Bangor University, Bangor, United Kingdom

\* seren.evans@bangor.ac.uk

**Data Availability Statement:** All relevant data are within the paper and its Supporting information files.

## Abstract

The cause of sport injuries are multifactorial and necessitate sophisticated statistical approaches for accurate identification of risk factors predisposing athletes to injury. Pattern recognition analyses have been adopted across sporting disciplines due to their ability to account for repeated measures and non-linear interactions of datasets, however there are limited examples of their use in injury risk prediction. This study incorporated two-years of rigorous monitoring of athletes with 1740 individual weekly data points across domains of training load, performance testing, musculoskeletal screening, and injury history parameters, to be one of the first to employ a pattern recognition approach to predict the risk factors of specific non-contact lower limb injuries in Rugby Union. Predictive models (injured vs. non-injured) were generated for non-contact lower limb, non-contact ankle, and severe non-contact injuries using Bayesian pattern recognition from a pool of 36 Senior Academy Rugby Union athletes. Predictors for non-contact lower limb injuries included dorsiflexion angle, adductor strength, and previous injury history (area under the receiver operating characteristic (ROC) = 0.70) Dorsiflexion angle parameters were also predictive of non-contact ankle injuries, along with slower sprint times, greater body mass, previous concussion, and previous ankle injury (ROC = 0.76). Predictors of severe non-contact lower limb injuries included greater differences in mean training load, slower sprint times, reduced hamstring and adductor strength, reduced dorsiflexion angle, greater perceived muscle soreness, and playing as a forward (ROC = 0.72). The identification of specific injury risk factors and useable thresholds for non-contact injury risk detection in sport holds great potential for coaches and medical staff to modify training prescriptions and inform injury prevention strategies, ultimately increasing player availability, a key indicator of team success.

**Funding:** The authors received no specific funding for this work.

**Competing interests:** The authors have declared that no competing interests exist.

## Introduction

The lower limb has the highest incidence of injury in Rugby Union [1–4], resulting in 50% of all days lost due to injury in international Rugby Union [5]. It is well established that as the burden of injury increases there are significant reductions in team performance [6]. Therefore, being able to identify when the risk of certain injuries are higher through modes of athlete monitoring could be invaluable in not only reducing the frequency of these injuries, but to inform injury prevention strategies and increase player availability, subsequently impacting team success [7]. Inevitably, the prediction of injuries has been investigated by applied sport scientists, who have presented a priori of reasons for collecting data on specific factors and utilizing linear based analyses [8]; undoubtedly this activity has been important in laying the foundations of existent sport injury literature. However, the explanatory power of these analyses may not infer the predictive power needed for sport injury risk detection [8–11]. As sport injuries are evidently complex and multifaceted [12], examining isolated risk factors associated with sport injury may be a reductionistic approach and unlikely to direct optimal injury prevention strategies [13, 14]. Despite the established evidence pointing towards the complexity of sport injury, there is limited research adopting a longitudinal, multifactorial approach in identifying predictive variables for injury whilst acknowledging the likelihood of non-linear interactions of variables, particularly with reference to specific anatomical locations. Deciphering which combination of variables contribute to increased injury risk by specific location or diagnosis (i.e. ankle injury, hamstring injury, non-contact lower limb injury), may aid in facilitating a more targeted approach to injury prevention.

Adopting a targeted approach to forecasting injuries means incorporating established and potential injury risk factors to accurately identify the combination of predictive variables for a specific injury. Performance based attributes such as greater maximal sprinting speed and aerobic fitness testing have been found to reduce the risk of lower limb injuries in sport [15, 16]. Furthermore, data derived from musculoskeletal screening have been identified as injury risk factors of the lower limb; for example ankle dorsiflexion angle deficits, and reduced adductor strength have previously been associated with an increased risk of lower extremity injuries such as patellar tendinopathy and adductor strains [17–20]. Monitoring athlete perceived ratings of muscle soreness is also an indicator of player training status and combined with factors of fatigue and musculoskeletal screening have the potential to identify greater likelihood of lower limb injury [21]. Some evidence also suggests that decreased eccentric hamstring strength is associated with greater risk of hamstring injury [22–24], with a well-established link between previous injury history and re-injury [25–29]. Despite the important role of linear analyses within early research in establishing injury risk factors, they may only demonstrate association between the variables and injury, and not their predictive importance [8, 10]. Therefore, a targeted pattern recognition approach to data handling in injury risk related research is pivotal for constructing robust injury modelling frameworks; allowing for the analysis of larger datasets including interactions and non-linear relationships of established modifiable and non-modifiable injury risk factors to formulate specific, interpretable injury risk profiles [8, 30].

Pattern recognition is a contemporary statistical approach whereby algorithms enable modelling of complex, multiple non-linear interactions between variables [31, 32]. These data handling methods overcome the limitations of typical logistic regression analyses previously used in "predicting" injuries in team sports, and further facilitate the development of rigorous, generalisable injury risk models inclusive of predictive variables. Emerging evidence highlight the efficacy of pattern recognition analyses in discerning predictive variables associated with heightened injury risk across various team sports [10, 32, 33]. Being able to effectively utilise

absolute and weekly change values of athlete monitoring components to communicate with stake holders which athletes are at an increased risk of injury could be invaluable for both coach education and athlete welfare. To our knowledge, there are no studies utilising a pattern recognition approach to identify lower limb non-contact injury risk factors in Rugby Union. Therefore, the aim of this study was to investigate whether a pattern recognition approach could be used to identify in-depth predictive variables that increase the risk of lower limb non-contact injuries in Rugby Union across multiple seasons based on data spanning training load parameters, performance testing results, previous injury history, and musculoskeletal screening. Pattern recognition algorithms will be used to create models comprising of the most predictive set of risk factors for lower limb non-contact injury, non-contact hip/groin injuries, non-contact hamstring injuries, non-contact knee injuries, non-contact ankle injuries and severe lower limb non-contact injuries.

## Methods

Data from thirty-six male regional Rugby Union players, representing the Senior Academy, were collected across two consecutive seasons (2018–2019, 2019–2020). Players attended three full days of training which included unit skills, speed and gym sessions lead by strength and conditioning and performance coaches, and two evening rugby-based sessions. League matches were played at the weekends from September until mid-May with full-time Physiotherapy provision and match-day doctors. Semi-professional Rugby Union is often seen as a key stepping-stone into professional sport and for player development, with players often being paid and some may train full-time whilst others will have primary full-time employment elsewhere [34]. Inclusion criteria for this study required all athletes to be in full-time training to ensure consistency across data points collected and to reduce the likelihood of missing data. All athletes were male semi-professional Rugby Union players and had to be over the age of 18. Each player provided written informed consent for the collection of: baseline anthropometric and performance testing; retrospective and prospective injury data; musculoskeletal screening; and training load parameters. For each individual player, 1740 in-season weekly data points were collected. Late season data was omitted due to there being numerous missing data points amidst the COVID-19 pandemic in the second season of this study. Player data was excluded if they had multiple missing data points across the season within measures training load, performance testing, musculoskeletal screening and injury history parameters. Part-time athletes were excluded from this study, and any athletes who left the team during the data collection period were also excluded. The study was approved by the Bangor University Ethics Committee of the School of Sport, Health and Exercise Sciences, which has more recently merged to form the School of Psychology and Sport Science (approval number: P20-18/19).

### Experimental approach

A two-year prospective observational design was implemented to identify noncontact lower limb injury risk factors in male senior academy Rugby Union players across two consecutive competitive seasons. Training load was quantified on each training and match-day, and further converted into Arbitrary Units AU) as: mean weekly training load; mean change in weekly training load; difference between highest and lowest weekly training load; and maximum change in weekly training load [35, 36]. Furthermore, performance testing data were recorded at the start of each season for: aerobic fitness [37]; sprint times [38]; 1 repetition maximum (1RM) absolute and relative strength in the back squat [39]; and lower limb power via a counter movement jump (CMJ) [40] which were implemented and collected by the Strength and Conditioning Coach during the preseason. Each athlete was further screened at the

beginning of the season for previous injury history including previous concussions by the Team Physiotherapist. Musculoskeletal screening was administered at the beginning of each training day including measures of: ankle dorsiflexion angle; hamstring strength; adductor strength; bodyweight; and perceived muscle soreness ratings [41–45]. These measures were conducted and recorded by the Primary Researcher and later extracted to the team's athlete monitoring database. Non-contact lower limb injuries were noted by binary classification (injury yes/no), as well as further categorization into their specific location and severity (hip/groin, knee, hamstring, ankle, more than 28 days lost due to injury). All raw data points measured were then collated and standardised into one Excel database for pattern recognition analysis, to further examine the best performing predictive variables from agreeing algorithms and identifying athletes with greater likelihood of sustaining lower limb non-contact injuries based upon the monitoring variables.

**Training load.** Training and match loads were quantified using the session Rating of Perceived Exertion (sRPE) method using a 10-point category ratio scale CR-10) and the session or match duration [46]. Thirty minutes post session; the player self-reported their perceived exertion using the Borg-CR10 scale for each individual session (for example; Unit Skills, Gym, Rugby Training, Conditioning) which was subsequently multiplied by the duration of the session (minutes), expressed in arbitrary units (AU) for each player [47, 48]. Weekly means of training load were collated for each player throughout each season, including absolute week-to-week changes in training load, maximum change in week-to-week load over the season, and pre-season cumulative training load (sum each athlete's training load in the preseason phase [49]).

**Performance testing.** Strength, speed, and aerobic fitness tests were performed during the pre-season phase to establish baseline performance. For this study, back squat 1RM was considered representative of lower limb strength and has previously identified as a predictive attribute for lower limb injuries [39]; athletes would perform a dynamic warm-up prior to testing, followed by utilizing an empty barbell to squat before gradually increasing the weight to 90% of their previous 1RM. Athletes were permitted three attempts from this threshold to establish their 1RM. Players were visually assessed by a UKSCA-certified strength and conditioning coach during each 1RM attempt to ensure appropriate technique was performed. Back squat 1RM absolute values were noted and then calculated relative to body mass for each player (1RM / body mass (kg)).

Player sprint speed was assessed over 10m and 40m from a stationery position. Athletes were permitted three attempts at maximum effort, with the fastest time being noted for analysis. Sprint speed was measured using a timing gate system (Brower Timing Systems, Utah, USA) using a single beam infrared light system with the data then sent directly to a handheld monitor. Inter-rater reliability of this procedure is considered at an inter-class correlation coefficient between 0.91–0.99 over 10m to 30m sprint tests [38].

The Bronco test [37] was administered to assess aerobic capacity and involved a shuttle-running time-trial over a total distance of 1200m. Distances of 0, 20, 40, and 60m were marked with cones and athletes were required to run from the 0-m to the 20-m marker and back, then run from the 0-m to the 40-m marker and back, and finally run from the 0-m to the 60-m marker and back. Completion of this sequence of shuttles was considered one repetition, with athletes completing five repetitions to cover 1200m as quickly as possible. Video-analysis was used to record the duration of each player's Bronco time. This test is commonly used in Rugby Union settings to measure maximal aerobic capacity and is considered a cost and time-efficient method for coaches to monitor high-intensity aerobic fitness [37].

A CMJ was used to establish lower limb explosive power, using a contact mat (Just Jump System, Probotics Inc., Huntsville, AL, USA). To measure jump height of a maximal effort jump from a static position; athletes would start on the contact mat in an upright position with

hands on hips followed by a dip into self-selected depth and would perform a vertical acceleration aiming to generate a maximal vertical jump height. During testing, athletes were permitted three attempts with 2-minute rest in-between each attempt, and the highest CMJ height was recorded for analysis.

**Injury history.** Previous injury history was noted as part of pre-season screening of the athletes and collected by the Team Physiotherapist. Injuries were further specified for each different injury location (groin, hamstring, knee, ankle) and previous injury history referred to a previous injury corresponding to the classified injury in the analysis. Additionally, if the athlete had sustained a concussion within the previous season, this was noted separately. Previous incidence of concussion has been found to increase the likelihood of subsequent soft-tissue injury in team sports [50, 51].

**Musculoskeletal screening.** Height (cm) and body mass (kg) were measured using a stadiometer (Bodycare Ltd, Warwickshire, UK) and digital platform scales (Model 705, Seca, Hamburg, Germany). Subjective measurements of muscle soreness for the lower limb were collected on a Likert scale of 1–10 prior to each training day [41]. A recording of 1 represented no soreness and 10 equalled very sore, and subjective markers have previously been deemed reliable and valid methods of recording muscle soreness [52]. Athletes would perform objective validated lower limb musculoskeletal screening tests prior to the commencement of each training day which would take place three times per week for the duration of the season, including: adductor squeeze (adductor strength; [42, 43]) dorsiflexion lunge angle with dominant to non-dominant leg comparison (ankle dorsiflexion range of movement; [44]) and hamstring squeeze with dominant to non-dominant leg comparison (hamstring strength; [45]). Detailed methods used for musculoskeletal screening are listed in S1 Table.

**Injury classifiers.** Players were assigned class membership on six binary classification variables: non-contact lower limb injury of mild to moderate severity (lower limb injured vs non-injured class), non-contact hip/groin injury (hip/groin injured vs non-injured class), non-contact hamstring injury (hamstring injury vs non-injured class), non-contact knee injury (knee injury vs non-injured class), non-contact ankle injury (ankle injury vs non-injured class), and severe non-contact injury (severe injury vs non-severe/non-injury class). Injury data was collated by a designated Physiotherapist for this study and was defined as non-contact lower limb injury resulting in time-loss from training and/or match-play within the data collection period [53]. If an injury resulted in time-loss within this time-frame, the following training load and monitoring markers were omitted from the study until the player returned to full-training.

## Statistical analysis

Bayesian pattern recognition analysis was performed collaboratively by RO and SE in R programming language (R Core Development, 2021), with the Tidyverse package used for data manipulation [54] and rWeka package used to compute WEKA machine learning algorithms [55, 56]. Bayesian pattern recognition analysis was performed a total of six times, once for each binary classification variable (non-contact lower limb injury, non-contact ankle injury, non-contact knee injury, non-contact hamstring injury, non-contact hip/groin injury and severe non-contact injury), to determine which of the 103 collected features (i.e., predictor variables such as eccentric hamstring strength discrepancy between limbs; see Table 1 for a full list) best predicted group membership (i.e., injured or non-injured). In line with previous work utilising pattern recognition in sport [56, 57], each instance of the Bayesian pattern recognition analyses comprised of three stages: pre-processing, feature selection, and assessment of classification accuracy.

Pre-processing was performed at the beginning of each pattern recognition analysis to standardize features and operationalise classification variables. Specifically, raw data of features

**Table 1. Potential predictive risk factor variable list.**

| Training Load | | | | |
|---|---|---|---|---|
| Mean Weekly Training Load | Mean Change in Weekly Training Load | Highest to Lowest Weekly Training Load | Maximum Increase in Weekly Training Load | Preseason Cumulative Load |
| **Performance Testing** | | | | |
| 10m Sprint | 40m Sprint | Bronco Fitness Test | Absolute Back Squat 1RM | Back Squat 1RM Relative to Bodyweight |
| Counter Movement Jump Test | | | | |
| **Injury History** | | | | |
| Previous Injury | Concussion in the Previous Season | | | |
| **Musculoskeletal Screening** | | | | |
| Bodyweight | Stature | Age | Playing Position | |
| Mean Weekly Muscle Soreness | Mean Change in Weekly Muscle Soreness | Highest to Lowest Weekly Muscle Soreness | Maximum Increase in Weekly Muscle Soreness | |
| Mean Weekly Adductor Squeeze Test | Mean Change in Weekly Adductor Squeeze Test | Highest to Lowest Weekly Adductor Squeeze Test | Maximum Decrease in Weekly Adductor Squeeze Test | |
| Mean Weekly Ankle Dorsiflexion Angle | Mean Change in Weekly Ankle Dorsiflexion Angle | Highest to Lowest Weekly Ankle Dorsiflexion Angle | Maximum Decrease in Weekly Ankle Dorsiflexion Angle | Left to Right Leg Difference in Weekly Ankle Dorsiflexion Angle |
| Mean Weekly Hamstring Strength | Mean Change in Weekly Hamstring Strength | Highest to Lowest Weekly Hamstring Strength | Maximum Decrease in Weekly Hamstring Strength | Left to Right Leg Difference in Weekly Hamstring Strength |

All 22 weekly variables were sub-divided into phases of the season [preseason, early, mid], resulting in 103 weekly variables entered into the feature selection stage.

was z-scored using their mean and standard deviation, prior to being converted into a vector from 0 to 100 (with a player's score of 50 representing a score equivalent to the mean and a score of 60 representing a score 1SD above the mean, etc). Data of classification variables (i.e., injury incidence) was coded in a way where 0 represented an absence of injury for a player during the season, whilst 1 represented the presence of an injury.

Feature selection was performed following pre-processing, for each instance of pattern recognition. This involved the use of correlation attribute evaluator [58], gain ratio attribute evaluator [59], relief F attribute evaluator [60], and info gain attribute evaluator [59] to identify the features which best predicted group membership within the classification variable (i.e., 0 or 1). Features which were identified as being among the top-20 strongest predictors of group membership by at least two feature selection algorithms formed part of a resultant feature 'model' which proceeded to the next stage of analysis. Note, the present top-20 and minimum of 2 agreeing algorithm criterions were set in-line with previous pattern recognition research in sport [56, 57].

The classification accuracy (i.e., accuracy with which injured/non-injured group membership is determined) of the models that emerged from the feature selection stage was assessed via naïve bayes [61], J48 decision tree [62], support vector machine [63], and K-nearest neighbour [64] classification algorithms. These analyses were iterated using a 'leave one out' cross-validation procedure [65]. The resultant output from these analyses provided a mean of the four classification algorithms ratings of classification accuracy (in percentage terms), sensitivity (a measure of correctly identified members of the non-injured group), specificity (a measure of correctly identified members of the injured group) and area under the receiver operating characteristic (ROC) (an overall measure of model performance when used for classification). The process of pattern recognition specifically for Rugby Union is schematically represented in Fig 1.

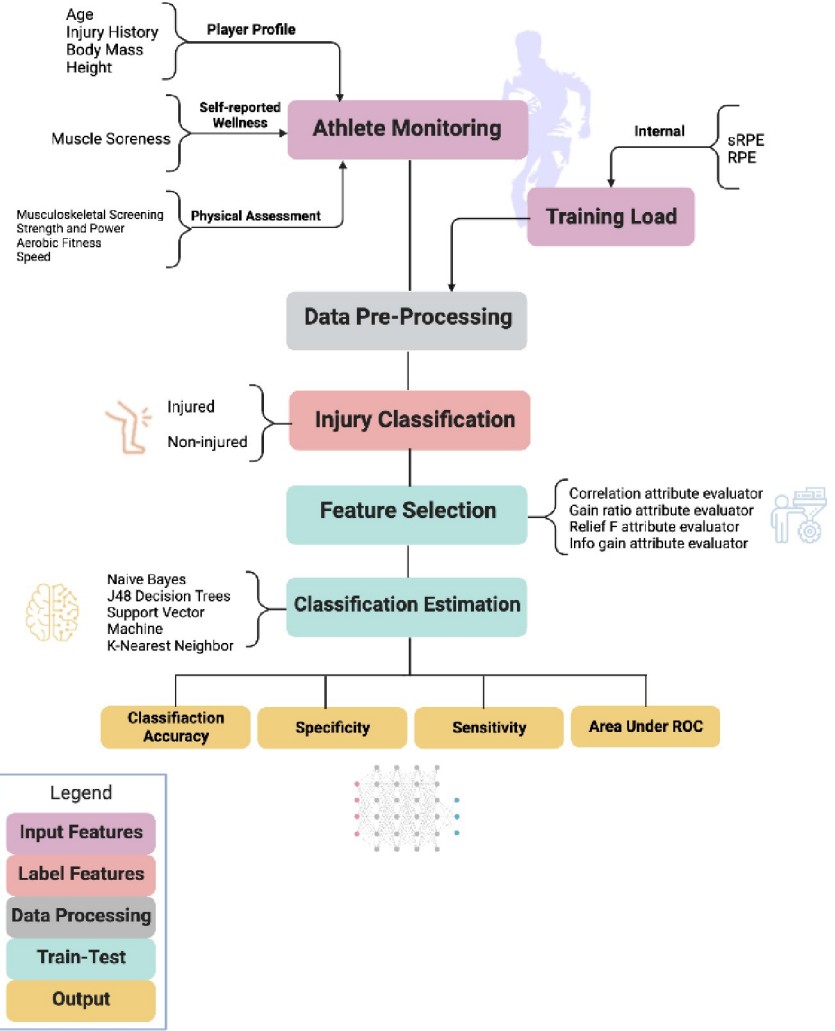

**Fig 1. An adapted Figure [10] demonstrating the process of injury prediction validation using a pattern recognition approach.** Athlete monitoring and training load refer to input variables for the pattern recognition algorithms. Injury classification is typically the binary response (injury yes/no) in labelling the training vectors. The models are those tested by the injury prediction algorithms in this study. Cross-validation and feature selection are processes of training, validating, and testing the models developed, further supported by the output determining the usefulness of the model. Furthermore, blue leaves describe how to train, validate, and test the model developed by the injury prediction algorithm.

## Results

Thirty-six athletes (Mean: age 20.7 ± 2.2 years, stature 183.3 ± 6.1cm, body mass 98.9 ± 11.5kg) participated in this study across two consecutive seasons (2018–19, 2019–20). Overall, there were 25 athletes who sustained lower limb non-contact injury (Mean: age 20.7 ± 2.2 years, stature 183.4 ± 6.6cm, body mass 100.1 ± 12.3kg), and 11 athletes not sustaining an injury (Mean: age 20.8 ± 2.4 years, stature 183.0 ± 5.3cm, body mass 96.1 ± 9.4). The most common mechanism of lower limb non-contact injury was running (16% of all non-contact lower limb injuries), and the most injured location was the ankle (52% of non-contact lower limb injuries), with the most common diagnosis being ankle sprains (48%). Table 2 contains the overall feature selection result, where models comprising of predictive risk factors were created for

Table 2. Algorithm agreements among feature selection models and their associated classification for non-contact lower limb injuries during the season.

| Number of feature selection algorithms in agreement | Models | | | | | | | | | | | |
|---|---|---|---|---|---|---|---|---|---|---|---|---|
| | Non-contact Lower Limb Injury | | | | Non-contact Ankle Injury | | | | Severe Non-Contact Lower Limb Injury | | | |
| | Training Load | Performance Testing | Injury History | Musculoskeletal Screening | Training Load | Performance Testing | Injury History | Musculoskeletal Screening | Training Load | Performance Testing | Injury History | Musculoskeletal Screening |
| 4 | | | | | | | | | | | | Early season mean change in muscle soreness; Position |
| 3 | | | | Mid-season maximum reduction in adductor strength | | | Previous Ankle Injury | | | | | |
| 2 | | | Previous Lower Limb Injury | Mid-season mean change in dorsiflexion angle | | 10m Sprint; 40m Sprint | Previous Concussion | Bodyweight; Preseason dorsiflexion angle; Early season maximum change in dorsiflexion angle | Highest to lowest difference in training load | 10m Sprint; 40m Sprint | | Early season mean change in adductor strength; Early season mean change in dorsiflexion angle; Early season mean change in hamstring strength |
| Area Under the ROC | 0.70 | | | | 0.72 | | | | 0.76 | | | |

**Table 3. Classification accuracy for severe non-contact lower limb injury during phases of the season.**

| Classifier | Classification Accuracy (%) | Sensitivity | Specificity | Area Under ROC Curve |
|---|---|---|---|---|
| Naïve Bayes | 88.88 | 0.93 | 0.78 | 0.93 |
| Support Vector Machine | 77.77 | 1.00 | 0.12 | 0.55 |
| K Nearest Neighbour | 75.00 | 1.00 | 0.00 | 0.78 |
| J48 Decision Tree | 86.11 | 0.93 | 0.67 | 0.77 |
| Mean | 81.92 | 0.96 | 0.39 | 0.76 |

Sensitivity = 1 –true positive for injured group. Specificity = 1 –true positive for non-injured group. ROC = Receiver operating characteristic.

**Table 4. Classification accuracy for non-contact lower limb injury during phases of the season.**

| Classifier | Classification Accuracy [%] | Sensitivity | Specificity | Area Under ROC Curve |
|---|---|---|---|---|
| Naïve Bayes | 75.00 | 0.46 | 0.88 | 0.76 |
| Support Vector Machine | 63.88 | 0.00 | 0.92 | 0.46 |
| K Nearest Neighbour | 72.22 | 0.19 | 0.96 | 0.77 |
| J48 Decision Tree | 83.33 | 0.55 | 0.96 | 0.79 |
| Mean | 73.6 | 0.30 | 0.93 | 0.70 |

Sensitivity = 1 –true positive for injured group. Specificity = 1 –true positive for non-injured group. ROC = Receiver operating characteristic.

different injury types. Injury models producing areas under the ROC of 0.7 and above are discussed, as areas under the ROC of between 0.7 to 0.8 represent fair to moderate diagnostic accuracy [66]. Models for non-contact hip/groin, non-contact hamstring, and non-contact knee injuries produced areas under the ROC ranging between 0.39 to 0.61, therefore were excluded from the results for discussion. Given the inherent complexity of sport injury, achieving greater values of diagnostic accuracy is difficult within injury prediction [8], however

**Table 5. Classification accuracy for non-contact ankle injury during phases of the season.**

| Classifier | Classification Accuracy (%) | Sensitivity | Specificity | Area Under ROC Curve |
|---|---|---|---|---|
| Naïve Bayes | 77.77 | 0.84 | 0.64 | 0.75 |
| Support Vector Machine | 72.22 | 0.88 | 0.37 | 0.62 |
| K Nearest Neighbour | 80.55 | 0.96 | 0.46 | 0.78 |
| J48 Decision Tree | 72.2 | 0.76 | 0.64 | 0.72 |
| Mean | 75.7 | 0.86 | 0.52 | 0.71 |

Sensitivity = 1 –true positive for injured group. Specificity = 1 –true positive for non-injured group. ROC = Receiver operating characteristic.

attention should be given to whether the predictive variables identified within the models are consistent with previous risk factor research and the applied implications of these findings. Pattern recognition analysis in this study identified models for severe non-contact lower limb injury, non-contact lower limb injury, and non-contact ankle injury with area under the ROC ranging between 0.70–0.76.

Table 3 shows a severe non-contact lower limb injury model from the classification analysis (Area under the ROC = 0.76). Players who sustained a severe non-contact injury saw less of a difference between their highest and lowest training load over the mid-season (2022.44 ± 397.36 AU vs 2406.59 ± 1266.86 AU), were slower in the 10m sprint (1.76 ± 0.14 s vs 1.69 ± 0.08 s), slower in the 40m sprint (5.44 ± 0.34 s vs 5.25 ± 0.29 s), saw greater muscle soreness increases in the early season (0.76 ± 1.24 vs 0.16 ± 0.56), greater reduction in adductor strength across the early season (-8.4 ± 15.65 mmHg vs -1.09 ± 3.27 mmhg), greater reduction in early season dorsiflexion angle (-1.13 ± 5.54˚ vs –0.07 ± 0.56˚), greater reduction in early season hamstring strength (-4.83 ± 17.54 mmHg vs –1.83 ± 4.99 mmHg) and were more likely to be forwards when compared to non-injured players.

Table 4 shows a non-contact lower limb injury model from the classification analysis (Area under the ROC = 0.70). Players who sustained a non-contact lower limb injury had a less reduction in adductor strength in the mid-season compared to non-injured players (-13.91 ± 10.78 mmHg vs -39.14 ± 31.05 mmhg), had a lower reduction in dorsiflexion angle in the mid-season compared to non-injured players (–0.23 ± 1.2˚ vs -0.70 ± 1.7˚) and had previously sustained a lower limb injury when compared to non-injured players.

Table 5 shows a non-contact ankle injury model from the classification analysis (Area under the ROC = 0.71). Players who sustained a non-contact ankle injury were slower in the 10m sprint (1.80 ± 0.15 s vs 1.69 ± 0.07 s), and the 40m sprint (5.43 ± 0.42 s vs 5.24 ± 0.23 s), had a lower dorsiflexion angle during the preseason (32.45 ± 11.06˚ vs 37.08 ± 15.82˚), greater reduction in dorsiflexion angle over the early season phase (-5.05 ± 2.98˚ vs –3.24 ± 1.93˚), and were heavier in bodyweight (105.7 ± 15.2 kg vs 95.9 ± 8.1 kg) as well as having sustained a concussion within the previous season and a previous ankle injury when compared to non-injured players.

## Discussion

The aim of this study was to identify the most predictive variables for non-contact lower limb injury in male Rugby Union players via a pattern recognition approach. We developed and evaluated predictive models comprised of features representing training load, performance testing, previous injury history and musculoskeletal screening markers collected throughout the phases of the season. Our findings not only successfully identify predictive models capturing the multifactorial nature of lower limb noncontact injury, but also explicitly specified thresholds for each of the features evaluated whereby there was an increased likelihood of injury; an invaluable tool for practitioners to utilise in implementing timely injury prevention strategies.

Severe non-contact lower limb injuries were attributed to smaller differences between highest to lowest midseason training loads, slower 10m and 40m sprint times, greater increase in muscle soreness in the early season, greater decrease in adductor strength in the early season, greater early season mean decrease in dorsiflexion angle, greater decrease in hamstring strength in the early season with forwards more likely to sustain an injury compared to backs. There is some evidence that a relationship exists between training load and injury risk in athletes [67–69], with underloading being a potential characteristic of predisposing athletes to non-contact injuries [70]. A smaller variation in weekly training loads within the mid-season

phase may suggest increased training monotony compared to the un-injured group, which may increase the risk of lower limb non-contact injury [71]. Additionally, enhanced sprinting ability has been considered a moderator for injury risk, and slower 10m and 40m sprint times increased injury risk in sub elite Rugby League players [72], reflective of our findings. Furthermore, isometric hamstring strength was measured in this study; reductions in hamstring strength was a predictor of severe lower limb non-contact injury, and previous studies do report that hamstring weakness predisposes athletes to injury [73, 74]. The severity of lower limb non-contact muscular strain injuries could be attributed to sudden activation of eccentric contractions of the hamstring muscles during the late swing phase of high-speed running [75–77]. Interventions for enhancing hamstring strength may therefore reduce lower limb injury severity in football and Rugby Union [78]. Tentatively from our findings, we may assume that greater training monotony, running exposure paired with limited sprinting ability and poor hamstring strength may increase the risk of severe lower limb non-contact injuries.

Self-reported measures of muscle soreness have been reported to be sensitive to changes in training load amongst team sports athletes [52], with our study indicating that greater increases in weekly perceived muscle soreness was symptomatic pre-lower limb injury. In recent machine learning approaches to injury risk within elite male volleyball, muscle soreness and training exertion predicted injury [79]. Additionally, reduced adductor strength and reduced dorsiflexion angle are associated with greater lower limb injury risk [18, 19, 80, 81], supporting these features as predictive variables for severe noncontact lower limb injury in this model. Restricted ankle dorsiflexion may also contribute to reduced hip and knee mobility, potentially increasing lower limb injury risk due to greater ground reaction forces through increased joint stiffness [18, 82]. Finally, we reported positional differences within the model with forwards at a greater risk of severe lower limb non-contact injury. Previous literature has stated trivial differences between injury severity for forwards versus backs [5, 83–85]. As the most common mechanism of non-contact lower limb injuries typically derive from running-related activities in Rugby Union [5, 85, 86], we can postulate that the greater risk for the forwards may be due to differing training stimulus compared with backs; backs are more likely to be involved in high-speed running activities in match-play and their training stimulus will replicate this [87–89], whereas the predominantly contact nature of the forwards position may lead to their game based training in preparation for high-speed running to be less developed [90]. In combination, the above features contribute to forwards with restricted strength and range of movement in their lower limb musculoskeletal screening, increased perception of muscle soreness, slower sprint times and greater mid-season training load monotony to have greater likelihood of sustaining severe non-contact lower limb injuries.

Non-contact lower limb injury risk factors included a lower change in maximum adductor strength in the mid-season, lower change in dorsiflexion angle in the mid-season mean and previous lower limb injury. Contrasting with previous findings in elite football that athletes with greater reductions in hip adductor strength were at an increased risk of lower limb injury [19, 80, 81]. As reduced absolute adductor strength has been found to increase injury risk in team sports, a possible explanation for our findings would be that the un-injured athletes in this analysis were 5% stronger than their injured counterparts at their preseason adductor strength baseline, potentially moderating their risk of lower limb non-contact injury. Similarly, it was identified that less reduction in ankle dorsiflexion was a risk factor for lower limb non-contact injury. As these results signify changes in the mid-season, it is important to consider natural decreases in flexibility over the course of typically congested match fixtures due to increased training loads and limited recovery [91]. Finally, previous lower limb injuries are considered primary risk factors for both recurrent injuries and new injuries in team sports [25–29], supporting the finding of this feature being a significant contributor to this injury

model. Despite some of these findings being atypical, it is important to note that models which are generated for specific diagnoses of lower extremity musculoskeletal injuries may be of greater sensitivity in multivariate modelling, producing more specific, interpretable findings for injury risk [92]. Future research within pattern recognition and injury risk should therefore be modelled on specific injury diagnosis to facilitate further specificity and generalisability of the models in an applied setting.

Non-contact ankle injuries were determined by lower mean dorsiflexion angle in the pre-season, reduced early season mean change in dorsiflexion angle, greater bodyweight, slower 10m and 40m sprint times, sustaining a concussion within the previous season, and history of previous ankle injury. Reductions in dorsiflexion angle is well established as having links to ankle and lower limb injury, by altering lower-extremity stiffness and forces generated through the ankle during landing mechanics [18]. Furthermore, heavier athletes in this study were at an increased risk of sustaining non-contact ankle injuries, with current evidence identifying greater body mass index as a predictor of ankle injury [8, 92]. Slower high-speed running ability was also identified as a predictor of ankle injury risk, and as previously discussed, reduced sprint speed has been associated with increased injury risk in team sports [16, 93]. Furthermore, the long-term effect of sustaining a concussion is being extensively researched in team sports, with some associating a previous concussion diagnosis to an increased risk of subsequent lower limb injury [50, 51], specifically lateral ankle sprains [94]. It is also important to consider that previous ankle sprains can result in significant reductions in ankle range of motion and chronic instability if not managed, with previous injury history being a strong risk factor for recurrence with regards to the clinical course of ankle sprains [95, 96]. Being a heavier athlete with a history of ankle and concussion injury combined with limited ankle dorsiflexion may not only increase joint loading, but also alter optimal movement patterns due to potential proprioceptive deficits following a concussion during activities such as running, landing, and cutting activities, creating this predisposition for re-injury in Rugby Union. By screening athletes in the preseason for ankle range of motion, sprinting ability and previous injury history, particularly previous concussion history, practitioners can implement timely and conservative interventions prior to season commencement to address discrepancies and reduce the risk of these preventable injuries.

Despite being able to identify injury prediction models for non-contact lower limb, non-contact ankle and severe non-contact injury, we were unable to substantiate injury models for knee, hamstring, or hip/groin injuries with adequate area under the ROC values of above 0.7. This is in line with previous literature stating that there were no identifiable risk factors for hamstring injuries in elite Australian Football using a similar statistical approach [97], however there is some evidence as to which variables could predict anterior cruciate ligament injuries in Basketball and Rugby [98, 99]. In future, incorporating more specific injury diagnosis could provide an opportunity for accurate injury prediction models to be formulated by pattern recognition algorithms, answering more complex questions beyond general injury risk [26].

## Strength and limitations

The acknowledgement of pattern recognition's increasing acceptance in sport injury research [92, 97–99] reflects the realisation that injuries result from intricate interactions amongst modifiable and non-modifiable risk factors within a multidimensional context [12]. Recognising this complexity of injury causation, this study moves away from linear approaches which may oversimplify the composite nature of injuries, as approaches such as logistic regression do not handle imbalanced datasets that are typically seen within injury research [32]. Innovatively, this study adopted a dual approach, considering both absolute values of features and tracking

weekly changes, thus aligning with previous literature outlining the significance of monitoring changes in repeated measured for a more nuanced understanding of injury risk [100]. The breadth of data collected within this study offers a unique insight into the multifaceted nature of non-contact lower limb injuries within Rugby Union.

There were no specific psychosocial variables utilized as features for the models. Despite the use of subjective perception of muscle soreness in this study, previous studies have emphasised the importance of psychosocial well-being and injury risk [101], with documented models stating that psychological stressors, coping resources, life event stressors, personality and emotional state may be precursors for injury outcomes [102]. Despite the absence of explicit psychosocial variables, the present dataset distinguishes itself due to its thoroughness in comparison to other studies when exploring injury risk factors in sport. The data points collected seamlessly complemented usual monitoring processes and did not cause any additional time commitments for athletes. This positions the study as a robust exploration of non-contact injuries, capturing a comprehensive range of physiological factors contributing to injury risk whilst incorporating streamlined monitoring systems. Future research should include monitoring datapoints for psychosocial features in order to formulate a more holistic, multidisciplinary approach to sport injury.

As there are some differences and similarities in the current literature as to what predicts musculoskeletal non-contact injury in sport, it is important to consider differences between key performance indicators across sports which may alter an athlete's injury risk. An example of this may be the links between lower extremity stiffness and flexibility with lower limb injury risk, with a suggestion that there is an "optimal" level of flexibility depending on the key performance indicators of each sport [103]. To overcome this, baseline models from similar sports should be used to compare model outputs to create more generalisable results, as it has been recommended that comparing behaviours of injury prediction models built from different sports should be analysed both collectively and separately [32]. While this study represents and inaugural exploration of predictive variables for injury risk in Rugby Union, a comparative analysis with common features across our three models could serve as a step towards future generalisability. However, due to this being the first study to our knowledge examining predictive variables for non-contact injury risk in Rugby Union, this is information was not available.

## Conclusion

This is the first study to utilize pattern recognition analysis to examine the injury risk factors which pose the greatest risk to Rugby Union athletes in sustaining lower limb non-contact injuries. The present study provides a comprehensive approach to injury risk, accounting for their multifactorial nature and non-linear behaviours. Our findings indicate that training load parameters, performance testing results, athlete previous injury history and musculoskeletal screening combined can create holistic models for non-contact lower limb injury risk. Noteworthy predictive variables included differences in training load, reduced strength across musculoskeletal markers of adductor and hamstring strength, altered ankle dorsiflexion angles, perceived muscle soreness, slower sprint times, playing position and previous injury history. Appropriate weekly training load management, screening for strength and range of movement discrepancies across the season, enhancing sprinting abilities, and addressing previous injury history inclusive of previous concussions are some key considerations recommended for practitioners in formulating injury prevention strategies for non-contact lower limb injuries.

## Supporting information

**S1 Table. Daily musculoskeletal screening procedures.**
(PDF)

## Author Contributions

**Conceptualization:** Seren Lois Evans, Julian Owen.

**Data curation:** Seren Lois Evans, Gareth Whittaker, Oran Elphinstone Davis.

**Formal analysis:** Seren Lois Evans, Robin Owen.

**Funding acquisition:** Seren Lois Evans, Eleri Sian Jones, James Hardy, Julian Owen.

**Investigation:** Seren Lois Evans, Gareth Whittaker, Oran Elphinstone Davis.

**Methodology:** Seren Lois Evans, Julian Owen.

**Project administration:** Seren Lois Evans, Julian Owen.

**Resources:** Seren Lois Evans, Gareth Whittaker, Oran Elphinstone Davis, Julian Owen.

**Software:** Seren Lois Evans, Robin Owen.

**Supervision:** Seren Lois Evans, Eleri Sian Jones, James Hardy, Julian Owen.

**Validation:** Seren Lois Evans, Julian Owen.

**Visualization:** Seren Lois Evans, Robin Owen, Julian Owen.

**Writing – original draft:** Seren Lois Evans, Robin Owen, Julian Owen.

**Writing – review & editing:** Seren Lois Evans, Robin Owen, Gareth Whittaker, Oran Elphinstone Davis, Eleri Sian Jones, James Hardy, Julian Owen.

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
