## [Decision Letter · Decision Letter 0]

10 Jun 2024

PONE-D-23-42941Non-contact lower limb injuries in Rugby Union: a two-year pattern recognition analysis of injury risk factors.PLOS ONE

Dear Dr. Evans,

Thank you for submitting your manuscript to PLOS ONE. After careful consideration, we feel that it has merit but does not fully meet PLOS ONE’s publication criteria as it currently stands. Therefore, we invite you to submit a revised version of the manuscript that addresses the points raised during the review process.

We look forward to receiving your revised manuscript.

Kind regards,

Holakoo Mohsenifar

Academic Editor

PLOS ONE

Journal Requirements:

**Additional Editor Comments:**

The manuscript was written very well and in details.

Reviewers' comments:

Reviewer's Responses to Questions

**Comments to the Author**

1. Is the manuscript technically sound, and do the data support the conclusions?

Reviewer #1: Yes

2. Has the statistical analysis been performed appropriately and rigorously? 

Reviewer #1: Yes

3. Have the authors made all data underlying the findings in their manuscript fully available?

Reviewer #1: Yes

4. Is the manuscript presented in an intelligible fashion and written in standard English?

Reviewer #1: Yes

5. Review Comments to the Author

Reviewer #1: The manuscript was written very well and in details. there are some points to be considered in the manuscript.

1- The ethical code number should be provided in the manuscript.

2- The inclusion and exclusion criteria are not clear. Write them clearly with related references.

3- Some sections are without references such as the experimental approach. why did you change the load weekly? please cite the references.

4- Were the experimental approach and all tests done by the same physiotherapist? and how about the analyses?

5- What kind of load did you use in this project?

6- How did you measure the strength of the hamstring and adductors?

7- The results for ROM were not provided and discussed in the manuscript. please consider them.

6. PLOS authors have the option to publish the peer review history of their article (what does this mean?). If published, this will include your full peer review and any attached files.

Reviewer #1: **Yes: **Nahid Rahmani

---

## [Author Response · Author response to Decision Letter 0]

28 Jun 2024

We thank the reviewer and editor for their time, and very constructive comments and suggestions, which in turn has certainly led to a stronger manuscript. For further details of our response to the comments made, please see attached Response to Reviewers letter, as well as the Revised Manuscript with Track Changes. Kind regards, Seren Evans.

---

## [Editor Report · Decision Letter 1]

4 Jul 2024

Non-contact lower limb injuries in Rugby Union: a two-year pattern recognition analysis of injury risk factors.

PONE-D-23-42941R1

Dear Dr. Seren Lois Evans,

We’re pleased to inform you that your manuscript has been judged scientifically suitable for publication and will be formally accepted for publication once it meets all outstanding technical requirements.

Kind regards,

Holakoo Mohsenifar

Academic Editor

PLOS ONE
---

## [Editor Report · Acceptance letter]

24 Jul 2024

PONE-D-23-42941R1 

PLOS ONE

Dear Dr. Evans, 

I'm pleased to inform you that your manuscript has been deemed suitable for publication in PLOS ONE. Congratulations! Your manuscript is now being handed over to our production team.

Kind regards, 

on behalf of

Dr. Holakoo Mohsenifar 

Academic Editor

PLOS ONE